# Imaging-Guided Percutaneous Puncture and Embolization of Visceral Pseudoaneurysms: Feasibility and Outcomes

**DOI:** 10.3390/jcm11112952

**Published:** 2022-05-24

**Authors:** Serena Carriero, Carolina Lanza, Pierpaolo Biondetti, Matteo Renzulli, Cristian Bonelli, Filippo Piacentino, Federico Fontana, Massimo Venturini, Gianpaolo Carrafiello, Anna Maria Ierardi

**Affiliations:** 1Postgraduate School of Radiodiagnostics, Università degli Studi di Milano, 20122 Milan, Italy; serena.carriero@unimi.it; 2Interventional Radiology Unit, Department of Radiology, Foundation IRCCS Ca’ Granda-Ospedale Maggiore Policlinico, 20122 Milan, Italy; pierpaolo.biondetti@unimi.it (P.B.); gianpaolo.carrafiello@unimi.it (G.C.); annamaria.ierardi@policlinico.mi.it (A.M.I.); 3Department of Health Science, Università degli Studi di Milano, 20122 Milan, Italy; 4Radiology Unit, Department of Experimental, Diagnostic and Speciality Medicine, Sant’Orsola Hospital, University of Bologna, 40138 Bologna, Italy; matteo.renzulli@unibo.it; 5Healthcare Professional Department, Foundation IRCSS Ca’ Granda-Ospedale Maggiore Policlinico, 20122 Milan, Italy; cristian.bonelli@policlinico.mi.it; 6Diagnostic and Interventional Radiology Unit, ASST Settelaghi, 21100 Varese, Italy; filippo.piacentino@asst-settelaghi.it (F.P.); federico.fontana@uninsubria.it (F.F.); massimo.venturini@uninsubria.it (M.V.); 7Department of Radiology, Insubria University, 21100 Varese, Italy

**Keywords:** visceral pseudoaneurysm, embolization, percutaneous approach, trans-arterial approach

## Abstract

Visceral artery pseudoaneurysms (VAPAs) are the most frequently diagnosed pseudoaneurysms (PSAs). PSAs can be asymptomatic or symptomatic. The aim of our study was to evaluate the safety and effectiveness of percutaneous embolization of VAPAs performed on patients with an unfeasible trans-arterial approach. Fifteen patients with fifteen visceral PSAs, with a median dimension of 21 mm (IQR 20–24 mm), were retrospectively analyzed. No patients were suitable for trans-arterial catheterization and therefore a percutaneous approach was chosen. During percutaneous treatments, two embolic agents were used, either N-butyl cyanoacrylate (NBCA) (Glubran II, GEM Milan, Italy) mixed with Lipiodol (Lipiodol, Guerbet, France) or thrombin. The outcomes of this study were technical success, primary clinical success, and secondary clinical success. In our population the 15 PSA were located as follows: 2 in the left gastric artery, 1 in the right gastric artery, 3 in the right hepatic artery, 2 in a jejunal artery, 1 in left colic artery branch, 1 in a right colic artery branch, 1 in the gastroepiploic artery, 1 in the dorsal pancreatic artery, 1 in an ileocolic artery branch, 1 in an iliac artery branch, and 1 in a sigmoid artery branch. 80% of PSAs (12/15) were treated with a NBCA:lipiodol mixture and 20% of PSAs (3/15) were treated with thrombin. Technical, primary, and secondary clinical successes were obtained in 100% of the cases. No harmful or life-threatening complications were observed. Minor complications were registered in 26.6% (4/15) of the patients. Percutaneous embolization of visceral PSA is a safe and effective treatment and should be considered as an option when the endovascular approach is unsuccessful or unfeasible.

## 1. Introduction

Pseudoaneurysms (PSAs) are clinical entities caused by a disruption in an arterial vessel wall, with blood collecting around the damaged artery and forming a sac that communicates with the artery and that is lined by media, adventitia, or tissues surrounding the artery [1,2]. The creation of a fragile lumen continuously filled by the blood flow can lead to very dangerous complications if not promptly diagnosed and effectively treated [3]. In fact, a sustained high arterial pressure on the elastic structures that are involved in the process can lead to a significant increase of the PSA volume. If not appropriately treated, PSA is a life-threatening condition and is associated with high morbidity and mortality due to complications such as complete rupture, deep vein thrombosis, infection, and compression of neurovascular structures [4]. 

The most common causes of this condition are localized inflammatory reactions, traumas, or penetrating injuries and invasive procedures (surgical or interventional) [5]. 

PSA can be asymptomatic, and therefore detected incidentally during radiologic examinations, or symptomatic [6]. Symptoms may be caused directly from the PSA (palpable thrill, pulsatile mass, audible bruit), may be secondary to its mass effect on adjacent structures (ischemia of surrounding tissues, neurologic symptoms, deep vein thrombosis), or to its rupture (shock, hematemesis, melena, retroperitoneal hemorrhage) [2,7,8].

PSA can be diagnosed with color-doppler ultrasound (CDU), contrast-enhanced ultrasound (CEUS), cross sectional examinations such as computed tomography (CT) and magnetic resonance (MR), or digital subtraction angiography (DSA) [6]. CDU has a sensitivity of 94–99% and a specificity of 94–97% in diagnosing PSAs [9]. The “yin-and-yang” sign is the typical US finding which is caused by the systolic and diastolic in and outflow of blood from the PSA [10].

Visceral artery PSAs (VAPAs) are the most frequently diagnosed PSAs [11] and are defined as abdominal PSAs involving the celiac trunk, the superior and inferior mesenteric arteries, or their branches. Clinically, they can present with gastrointestinal or abdominal bleedings and/or compression of nerves and other adjacent structures, especially when reaching a considerable volume such as in advanced stages [6].

The first therapeutic approach for PSA has been surgery. However, in the last few years, interventional radiology (IR) approaches have become increasingly used and sometimes preferred to a classic open surgical operation due to their lower invasiveness and complications rates [12]. 

To date, the most desirable interventional treatment is a classic endovascular exclusion of the PSA from circulation, which is achievable with coils, stents, and injectable liquids. Various techniques are possible, such as embolization (including the “sandwich” or “sac-packing” technique), flow exclusion through stent delivery, and stent-assisted coil embolization [6,13].

Nevertheless, in patients with failed or impossible trans-arterial approach, percutaneous embolization with glue, thrombin, gelfoam or coils has shown to be an effective and valid alternative [14,15,16,17].

This study aims to report our experience in percutaneous PSA embolization performed on patients with an impossible trans-arterial approach, in particular reporting clinical success and complication rates (during procedure and at follow up) as measures of safety and the effectiveness.

## 2. Materials and Methods

### 2.1. Patients

For this retrospective observational study, conducted in accordance with the Declaration of Helsinki, all patients with PSAs and an impossible trans-arterial approach who were treated with percutaneous embolization at our Institution in the last 10 years were retrospectively reviewed. In all cases, the diagnosis was made by multidetector CT using a 64 slice CT scanner (Philips, Amsterdam, The Netherlands). Trans-arterial embolization was tried in all cases and judged unfeasible due to very small caliber of the feeding PSA artery, unsatisfactory catheterization of the feeding artery and/or inability to catheterize the “exit” artery beyond the PSA, or the presence of multiple collaterals distal to the PSA.

Written informed consent was obtained from each patient or their relatives before embolization. Internal Review Board approved the retrospective review of the data.

### 2.2. Inclusion and Exclusion Criteria

Patients included in the study presented with severe anemization (hemoglobin drop > 2 g/dL in the last 24 h). In their recent history, they had undergone abdomino-pelvic surgery, percutaneous procedures, or had suffered from abdominal infections or traumatic events (downfall, motorbike or car crash). They were hemodynamically stable or had reached stability through resuscitation maneuvers. All patients selected for percutaneous approach had PSAs arising from peripheral arterial branches and not from main visceral arteries (ie segmental arteries were treated within the liver).

Uncorrectable coagulopathy was an absolute contraindication to the procedure. 

In addition, among the contraindications to percutaneous puncture embolization for visceral PSAs, there are unfavorable anatomical location for percutaneous needle placement (higher risk to hollow viscera and other major vessels), unfavorable neck-to-dome ratio (>1), demonstration of high flow arteriovenous fistulous communication. 

### 2.3. Transarterial and Percutaneous Procedure Technique and Follow-Up

Procedures were performed on an angiographic table (Philips Azurion 7 B20/15, Philips, Best, Amsterdam, The Netherlands) by two interventional radiologists with more than 10-year experience in endovascular and percutaneous techniques. 

All patients were treated under moderate sedation and local anesthesia at the puncture site with lidocaine was always performed at the beginning of the procedure. 

Arterial access was obtained in all patients through the femoral route using a 5 Fr vascular access sheath. Mesenteric arteries or celiac trunk were catheterized using Cobra C1 or Simons 1 catheter (Cordis, Miami Lakes, FL, USA). Angiograms and enhanced cone-beam CT (CBCT) examinations were performed to individuate the PSAs and to plan the route to reach it. A 2.7 Fr Progreat microcatheter (Terumo Inc., Tokyo, Japan) was used to super-selectively access the target arteries. 

In all of the patients of our series, trans-arterial catheterization was impossible for the anatomical characteristics mentioned above.

The PSA was identified by US examination and/or at fluoroscopy as pooling of contrast media whin its lumen. Percutaneous access was obtained percutaneously with a 22-gauge Chiba needle (Cook Incorporation, Bloomington, Indiana, USA). under US and fluoroscopic guidance; sometimes multiple angiogram studies were performed in different projextions to optimize the needle progression. Before any injection, the needle position correctness was confirmed with a sacculography.

Two embolic agents were used, either a mixture of N-butyl cyanoacrylate (NBCA) (Glubran II, GEM Viareggio, Italy) and lipiodol (Lipiodol, Guerbet, Villepinte, France), or thrombin. The NBCA:Lipiodol rato used was 1:2 and 0.2–0.6 mL of the liquid mixture were injected once or repeatedly if needed (Figure 1 and Figure 2).

Prior to using NBCA glue, the Chiba needle was repeatedly washed with a 5% dextrose solution; the glue was then injected under fluoroscopic guidance until the PSA was judged filled or small extravasation of glue was observed. 

Three hundred units of thrombin (1000 units/mL) were injected through the needle under combined guidance (US and fluoroscopic) for patients treated with thrombin. Regardless of the embolic agent used, after completing embolization the Chiba needle was removed and confirmation of complete PSA exclusion from circulation was demonstrated by a final US exam and/or angiogram.

### 2.4. Outcomes

Technical success was defined as complete embolization of PSA observed at the end of the procedure. Primary clinical success was defined as stabilization of the patient’s vital parameters and hemoglobin values stability at short-term follow-up (5–6 days after procedure). Secondary clinical success was defined as the absence of re-bleeding during the follow-up period (12–60 months), which included CECT evaluations at 1, 3, 6, 12 months, and every 12 months for a maximum of 60 months. Major and minor complications were recorded and classified following the Society of Interventional Radiology’s classification system [18].

## 3. Results

A total of 15 patients, 9 men and 6 women, with a median age of 49 years old (IQR 38–55 years), were included in this study. Three patients developed abdominal PSA following pancreatitis, 2 after biliary percutaneous procedures, 6 due to abdominal abscesses, 2 after surgery, 1 after a TIPS attempt and 1 after abdominal trauma (Table 1).

CECT identified 15 PSA in our population localized as follows: 2 in the left gastric artery, 1 in the right gastric artery, 3 in the right hepatic artery, 2 in the jejunal artery, 1 in a branch of the left colic artery, 1 in a branch of the right colic artery, 1 in the gastroepiploic artery, 1 in the dorsal pancreatic artery, 1 in a branch of the ileocolic artery, 1 in ileac artery branch, 1 in a sigmoid artery branch (Table 1).

The median dimension of PSA was 21 mm (IQR 20–24 mm) with a minimum diameter of 18 mm and a maximum diameter of 30 mm.

80% of PSA (12/15) were treated with a NBCA: lipiodol mixture and 20% of PSA (3/15) with thrombin.

Technical success rate was obtained in all of the patients. Primary clinical success was achieved in all of the patients: all patients achieved complete hemostasis following embolization, none required repetition of the procedure and no blood transfusion were needed during in hospital stay after procedure.

No harmful or life-threatening complications were observed during the procedures. Minor complications (the asymptomatic migration of the embolic agent) were registered in 26.6% (4/15) of the patients.

Secondary clinical success was obtained in all patients, meaning that no patient presented signs of re-bleeding during follow-up (12–60 months); four patients died during follow up for other reasons (2 cancer progression, 1 cerebrovascular acute event and 1 acute myocardial infarction, after 30, 45, 24, and 54 months, respectively) and 1 patient was lost after 48 months of follow up. Of the remaining patients, nine are all still alive and none reported further bleeding episodes or procedure-related complications.

## 4. Discussion

PSA is due to the disruption of the artery wall [19]. Artery wall trauma can be caused by a direct iatrogenic injury (surgery, endovascular procedures) or an indirect injury mechanism (infection, perivascular inflammation) [20]. In this study, both etiopathogeneses are included with nine patients having developed PSA through an indirect injury (pancreatitis, abdominal abscess) and six after a direct iatrogenic trauma (surgery, endovascular procedures).

There is a high risk of spontaneous PSA rupture due to the pulsatile blood flow directed to and from the PSA through its neck, which can cause a rapid PSA enlargement and in the worse cases a rupture [21]. The risk of rupture is more frequent in the hepatic artery (80%), pancreatic arteries (75%), and superior mesenteric artery (38%) [22].

Furthermore, PSAs are associated with a high mortality rate [23]. In this scenario, prompt intervention is a cornerstone. In past years PSAs were treated through surgical repair, which is invasive and associated with high morbidity and mortality rates [2,24].

Nowadays, mini-invasive interventional radiology treatments are available which can be performed by endovascular and/or percutaneous approach [14]. Compared to surgical therapy, radiologic treatments have advantages such as the minimally-invasive nature, which implies lower morbidity rates and faster recovery times, and the possibility to treat PSAs that are surgically challenging.

Whenever technically possible, the endovascular approach is preferred. In literature endovascular embolization success rates range from 75 to 100% and morbidity rates range from 14 to 25% [25,26].

Percutaneous embolization, usually used for the treatment of femoral arteries’ PSAs, is chosen when endovascular approach is not feasible [27]; various imaging modalities can be used as guidance during percutaneous treatment (including US, CT and fluoroscopy), and their use in combination has also been described [14].

In this study, we considered percutaneous embolization of visceral PSAs only in patients in which an endovascular approach was not feasible, especially when the artery of origin of the PSA was inaccessible and/or its occlusion was not a viable option [17,26,28]. Furthermore, the PSAs we considered for percutaneous treatment were those involving peripheral branches, while percutaneous approach was not considered in those cases arising from the main arteries for the significantly higher complication risk. In our experience clinical and technical success rate were 100%, without any case of recanalization. We treated segmental arteries and not main hepatic artery or visceral branches to re-duce the accidental spread of thrombogenic material in non-target districts with ischemic complications. 

Moreover, as already described in literature, the percutaneous embolization approach is a feasible option since PSAs, unlike true aneurysms, have a very thick and fibrous wall which reduces the risk of rupture after puncture [12].

In this series good results were achieved using two different liquid embolic agents, glue, and thrombin. Satisfactory technical success and complication rates using these agents have been described by other authors as well [29,30]. Moreover, the use of different embolic agents such as fibrin and collagen has also been described, representing a valid alternative [31].

Technically, we injected the embolic agents into the PSA sac in all cases, but Kenkichi et al. showed that whenever puncturing the sac is difficult due to size or location, injection of thrombin even into the collection of fluid surrounding the PSA can be a good alternative [14].

The minor complications rate observed in this study was low and there were no consequences. However, in literature complications such as vessel thrombosis or infection, false puncture of PSA are also reported [32].

The limitations of this study are its retrospective nature, such as the limited sample size and the heterogeneity of the series in terms of etiology.

On the other hand, to the best of our knowledge, the presented sample size, composed of 15 patients, is the most numerous present in the available literature.

## 5. Conclusions

In conclusion, this study shows that percutaneous embolization of visceral PSA is a safe and effective treatment and should be considered as treatment option when the endovascular approach is unsuccessful or unfeasible.

## Figures and Tables

**Figure 1 jcm-11-02952-f001:**
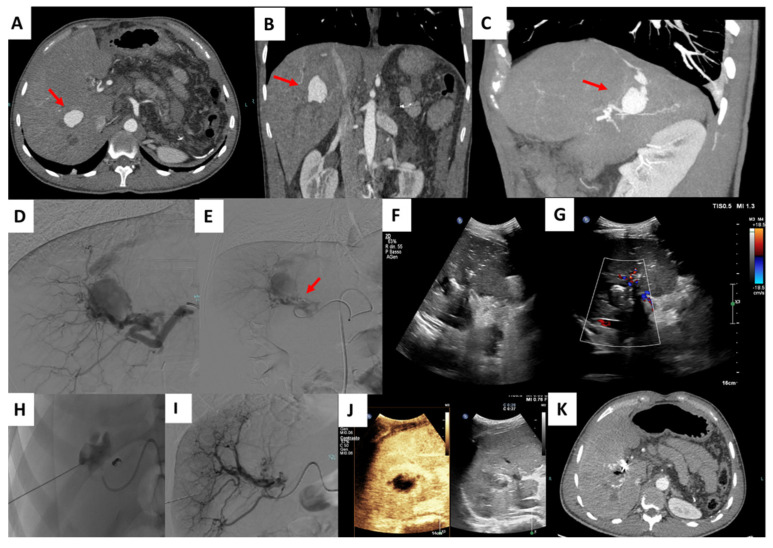
Percutaneous treatment of an intra-parenchymal PSA. Contrast enhancement CT (CECT) shows a right intrahepatic PSA: Axial (**A**), coronal (**B**) and MIP reconstruction (**C**). Selective right hepatic artery angiogram confirmed the presence of the PSA (**D**). After proximal embolization with microcoils (arrow), the angiogram demonstrated the persistence of the PSA (**E**). Percutaneous US guided puncture was performed (**F**,**G**). Saccography confirmed the correct position of the needle (**H**). Final angiogram demonstrated the complete exclusion of the PSA (**I**). Immediately after procedural contrast enhancement US (CEUS) confirmed the complete embolization of the PSA (**J**). CECT performed during follow up further confirmed exclusion of the PSA (**K**).

**Figure 2 jcm-11-02952-f002:**
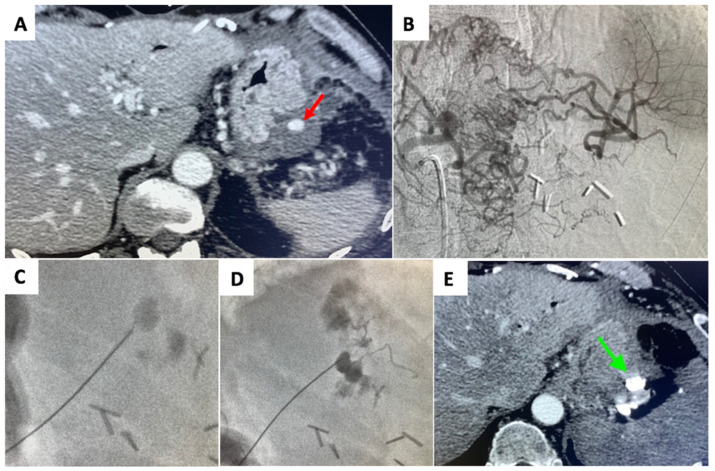
Percutaneous treatment of a left gastric artery PSA (**A**) Contrast enhanced CT demonstrated the PSA (arrow). (**B**) Selective angiography of celiac trunk demonstrated unfeasibility of an endovascular approach due to impossibility to reach the PSA (**C**) Under fluoroscopic guidance, percutaneous puncture of the PSA (**D**) and its embolization with N-butyl cyanoacrylate (NBCA) (Glubran II, GEM Italy) were performed. (**E**) Contrast enhanced CT scan after the procedure, showed the complete occlusion of the PSA (arrow).

**Table 1 jcm-11-02952-t001:** Clinical data of patients with PSA embolized with percutaneous treatment.

Patient Sex/Age	Cause	Arterial Territory Involved	PSA mm	Needle, Embolic Agent	Complications
M 49	pancreatitis	left gastric a	20 mm	22G, glue	asymptomatic splenic migration
M 54	biliary operation	right hepatic a	21 mm	22G, glue	no
M 25	TIPS	right hepatic a	30 mm	22G, thrombin	no
M 22	biliary operation	right gastric a	25 mm	22G, glue	asymptomatic duodenal migration
M 58	surgery	digiunal a	22 mm	22G, glue	no
F 45	abdominal abscess	branch of left colic a	24 mm	22G, thrombin	no
M 38	abdominal abscess	branch of right colic a	20 mm	22G, glue	no
F 60	pancreatitis	left gastric a	18 mm	22G, glue	no
F 52	abdominal abscess	right hepatic a	24 mm	22G, glue	no
M 48	pancreatitis	gastroepiploic aa	20 mm	22G, glue	no
M 54	surgery	dorsal pancreatic a	23 mm	22G, glue	no
F 28	trauma	first jejunal a	22 mm	22G, glue	asymptomatic migration
M 55	abdominal abscess	branch of the ileocolic a	20 mm	22G, glue	no
F 49	abdominal abscess	ileal branches	18 mm	22G, thrombin	no
F 58	abdominal abscess	sigmoid branch of IMA	20 mm	22G, glue	asymptomatic migration

## Data Availability

The data presented in this study are available on request from the corresponding author.

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
