# Peer review of "Imaging-Guided Percutaneous Puncture and Embolization of Visceral Pseudoaneurysms: Feasibility and Outcomes"

_jcm, 2022, doi:10.3390/jcm11112952_

Round 1

Reviewer 1 Report

The ethical aspect for the retrospective study isn't correctly explained. The patient or relative consent was obtained for the intervention, not for the publication of data collection thus this sentence is to be deleted. An IRB is probably required to re-use data for the purpose of publication and also alived patient's information.

Complications are collected but the timeline and type of last follow-up isn't clear. Procedures from last ten years were used for the report but last FU is 5 years. Vital status at the time of the data collection should be provided.

Reviewer 2 Report

The authors of the current manuscript make a retrospective analysis of 15 patients with visceral artery pseudoaneurysms (VAPAs), who underwent percutaneous embolization, aiming to evaluate the safety and effectiveness of this procedure. The topic of the article is original and the presented material is interesting from clinical point of view, since there are no large clinical studies in this sphere. Most of the available data is based on case reports, analysis of case series and small retrospective or prospective, open-label, non-randomized clinical studies. Any results from a new study about VAPAs management would be useful to clinicians, particularly if therapeutic options are limited, as for the study population, described by the authors (patients, not suitable for trans-arterial catheterization). According to the authors of the manuscript, the procedures applied for treatment of their study population achieved primary and secondary clinical success in 100%, which is very impressive, particularly because these results were obtained at the cost of just minor complications.

In terms of the technical quality of the manuscript: The paper is well written – the authors have observed the rules for writing a scientific paper for a specialized medical journal. The text is clear and easy to read. The conclusions are consistent with the evidence and the arguments presented in the main text. They address the main question that has been posed.

The main limitations of this paper (admitted also by the very authors) are: 1. The small sample size of the study population that does not allow extrapolation of the results to all patients with VAPAs and similar clinical profile; 2. The retrospective character of the analysis – a prospective study would allow a follow-up that would reveal the long-term effect of percutaneous embolization in VAPAs patients and the conclusions would be more persuasive.

These limitations however do not belittle the clinical and scientific assets of the manuscript. I recommend it to be considered by the Editors in its current version.

Round 2

Reviewer 1 Report

Previous questions were appropriatly addressed. 

Author Response

Thank you for your time. Your comments have helped us significantly improve the paper.